# *Culicoides* (Diptera: Ceratopogonidae) Abundance Is Influenced by Livestock Host Species and Distance to Hosts at the Micro Landscape Scale

**DOI:** 10.3390/insects14070637

**Published:** 2023-07-14

**Authors:** Armin R. W. Elbers, José L. Gonzales

**Affiliations:** Department of Epidemiology, Bioinformatics and Animal Studies, Wageningen Bioveterinary Research, 8200 AB Lelystad, The Netherlands; jose.gonzales@wur.nl

**Keywords:** cow, sheep, kernel, attraction, black-light suction trap, distance-dependency

## Abstract

**Simple Summary:**

*Culicoides* abundance is a critical component for transmission modelling of vector-borne diseases. It can be anticipated that *Culicoides* abundance is not homogeneously distributed in the landscape. Modelling parameter estimates are mostly extrapolated from indirect measurements using the most commonly used method worldwide: the light-trap (LT). However, it would be difficult to extrapolate *Culicoides* LT catches for modelling purposes due to the inherent biases of LT catches. The use of a sweep-net and performing collections during the hours before nightfall, when the LT is ineffective, can circumvent this bias. The aim of our study was to investigate the influence of baits (dairy cow, sheep, and LT) and the distance of measurement to hosts on *Culicoides* abundance using sweep-net catches. Our results indicate that cows are a far stronger attractor of *Culicoides* midges than sheep, and that both livestock hosts are far stronger attractors of *Culicoides* midges than LTs. *Culicoides* abundance declined significantly with increasing distance from hosts; this phenomenon was much stronger for dairy cows than for ewes. In contrast, *Culicoides* abundance increased with increasing distance from the LT, pin-pointing the apparent shortcomings of the LT as a surrogate host to lure midges.

**Abstract:**

The vector/host ratio and host preference are important parameters for the modelling of vector-borne livestock diseases. It can be anticipated that *Culicoides* abundance is not homogeneously distributed in the landscape. We investigated the influence of host species (dairy cow, sheep, and a light-trap (LT) as a surrogate host) and distance of measurement to hosts on *Culicoides* abundance using a randomized block-design with 12 measuring days and seven 3-min aerial sweep-netting sessions per whole hour at three distances to the host (0, 10, and 25 m), from five hours before to and including one hour after sunset. Dairy cows were found to be a far stronger attractor of *Culicoides* midges than sheep, while both hosts were far stronger attractors of midges than the LT. *Culicoides* abundance declined significantly with increasing distance from the livestock hosts; this phenomenon was much stronger for dairy cows than for ewes. In contrast, *Culicoides* abundance increased with increasing distance from the LT, pin-pointing the apparent shortcomings of the LT as a surrogate host to lure midges. Our data indicate that livestock host species and the distance from these hosts have a profound effect on *Culicoides* abundance in the landscape.

## 1. Introduction

The vector/host ratio and host preference are important parameters for the modelling of vector-borne livestock diseases, but there are only sparse field data to use for estimates with which to parameterize vector–host disease transmission models [1,2,3]. Furthermore, parameter estimates are mostly extrapolated from indirect measurements using the most commonly used method worldwide—the black-light suction trap (LT)—for catching midges [4]. By experience, it was established that the presence of livestock in the vicinity of the suction light trap increases the number of *Culicoides* trap catches [5,6], which is an indication that animal hosts act as attractants to midges. Currently, *Culicoides* biting midges are captured using several methods, including various models of the LT, the truck-trap, the mouth-aspirator (pooter) and mechanical aspirator, the drop-tent bait-trap, and the sweep-net [7]. Depending on the research questions to be answered, an appropriate collection method should be selected.

The LT enjoys advantages over other methods due to it being both labor inexpensive and easy to operate. In addition, because of the worldwide application of the LT to ascertain *Culicoides* presence and abundance, LT data are used to model the spread of vector-borne diseases [4]. However, field studies [8,9] indicate that the LT is not well suited to obtaining information that pertains more directly to the vertebrate host, such as accurate estimates of the vector/host ratio, attack and biting rates, and host preference. Furthermore, it has been established that the LT is not very good at attracting midges by itself [8,10], and any attraction of *Culicoides* midges by the LT only happens if the LT is placed at a very short distance from an animal host [10]. In addition, *Culicoides* species distribution estimates are biased when they are based on LT data due to the fact that the LT becomes effective only after nightfall, causing the abundance of the largely nocturnal *C. obsoletus* complex in the LT to invariably supersede that of the more diurnal *C. chiopterus* and *C. dewulfi* [9,10]. The same phenomenon has been observed for mosquito species after comparing sweep-net with light trap catches [11].

Parker [12] was one of the first to use aerial sweep-netting to catch *Culicoides* midges as a means to avoid biases in midge abundance data during the part of the day when the light intensity is too high for a LT to attract *Culicoides* midges. There is good correlation between direct aspiration (pootering) and sweep-netting results with respect to *Culicoides* midge distribution [9].

It can be anticipated that *Culicoides* abundance under field conditions is not homogeneously distributed in the landscape, and that it will be difficult to extrapolate *Culicoides* LT catches under natural field conditions for modelling purposes due to the aforementioned inherent biases. Lueken and Kiel [13] and Rigot et al. [14] showed that there was a slight but consistent decrease in the *Culicoides* abundance collected by LT in pastures surrounding livestock farms with increasing distance from the farms, which is a further indication that *Culicoides* midges are not homogeneously distributed in the landscape and that *Culicoides* presence is higher close to animal housing.

The purpose of our study was to examine *Culicoides* abundance, as measured by sweep-net, in order to determine the influence of livestock host species and distance to hosts at the micro landscape scale.

## 2. Materials and Methods

### 2.1. Study Site

The study was conducted at a dairy farm in the municipality of Ermelo, eastern Netherlands, in the May–June period (geographical location: 52°18′ N, 5°43′ E). The pasture used for the study measured 2.28 ha (approximately 120 m × 190 m). Three sites on a pasture, forming a rough triangle with about equal distances between the sites, were selected for the experiment and were situated approximately 150–200 m from the milking shed where 65 dairy cows were housed each night. Additionally, two Texelaar ewes were kept on a small pasture situated 50 m from the milking shed. The three experimental sites were separated from each other by a distance of approximately 80 m to avoid any one site exerting influence over any other during the trapping sessions. At each site, the following baits were used as attractants of *Culicoides* midges: a seven year-old dairy cow (100% Holstein-Friesian, black and white; approximate weight: 750 kg), a two year-old ewe (Texelaar sheep, white, shorn two weeks before the study started; approximate weight: 35 kg) and an Onderstepoort Veterinary Institute (OVI) black-light suction trap (LT) with an 8 W black light tube [5]. None of the animals were treated with insecticides before or during the experiment.

### 2.2. Study Design

The peak of *Culicoides* midge activity occurs around sunset [8,15]. Hence, approximately 6 h before sunset (SS − 6 h), the baits were installed at their designated positions in the pastures. The LT was installed on a tripod, 2 m above ground level. Each of the two animal hosts and the LT were randomly assigned to a site. In total, 12 replicate measurements (days) were made. The number of replicates used in our study is comparable with those of other *Culicoides* field experiments [8,16,17,18,19] aimed at testing biologically relevant differences. Daily observations were made only when the following weather conditions applied: windless, no rain, and temperatures > 10 °C; all collection days were bright, at most only partly cloudy. Sunset (SS) occurred at around 22:00 h, and times varied by only a few minutes throughout the study (Central European Summer Time). In each hour from SS − 5 h to and including SS + 1 h, *Culicoides* midges were sweep-netted at three different distances from the baits: 0 m, 10 m, and 25 m. An overview of the study design is shown in Figure 1.

For this, circles of 10 m and 25 m distance to the baits were constructed by setting the circles with white-colored wooden pegs in the pasture ground. For each whole hour on an experimental day, sweep-netting was randomly assigned to a distance; there was a down-time period of 5 min between the end of a sweep-net session at one distance and the start of sweep-netting at another distance. Sweeping involved the use of a commercially available lightweight gauze insect butterfly sweep-net (diameter 46 cm; aerial bag was made of closely woven off-white Dacron chiffon with mesh size: 70 × 70 per inch; insect net #7318WA with an alloy handle 2 m in length, BioQuip Products Inc. (Compton, CA, USA). Aerial sweep-netting was standardized with figure-8 sweeps at a calm sweeping pace (360 degrees; going from front to behind the sweeper) for 3 min while the collector was walking continuously circling the animals, which were tethered during the collection period, or the LT at the predetermined distance, as set out with wooden pegs. The catching results around the animal hosts with the sweep-net consisted of a combination of midges flying toward the host in order to take a blood meal (attraction), midges flying away from the host after having taken a blood meal, and those that were not interested in the host. However, the sweep-net result at 0 m of the LT merely provided a representation of midges flying toward the LT, because the LT traps the midges that come close to it (i.e., they are forced by a ventilator into a beaker containing soap hanging under the device). Therefore, for a fair comparison, it was deemed reasonable to add 5% of the LT catch to the values associated with the sweep-net at 0 m around the LT. Sweep-netting was done for 3 min each whole hour, while the LT operates for 60 min each hour; with this in mind, we determined that 5% (=3 out of 60 min) of the trapping result of the LT trap should be taken into account.

### 2.3. Culicoides Identification

*Culicoides* midges were stored in 70% ethanol until they were counted. *Culicoides* midges were identified morphologically primarily using the works of Campbell and Pelham-Clinton [20], Glukhova [21], and Delécolle [22]. All *Culicoides* midges collected in the study were counted, and no subsampling was done. The *C. obsoletus* complex comprised two species: *C. obsoletus* and *C. scoticus*; *C. chiopterus* and *C. dewulfi* are not deemed to form part of the *C. obsoletus* complex and so are treated separately [23]. Female midges were age-graded as either nulliparous, pigmented parous, gravid, or freshly blood-fed (engorged) [24].

### 2.4. Statistical Analysis

The total numbers of *Culicoides* or blood-fed *Culicoides* per observation day collected by sweep-net during the sampling period were estimated by calculating the area under the curve (AUC), which is a function of the number of *Culicoides* sweep-netted at each specific time of sampling and the number of times sampling was done per day. Since we were dealing with counts, AUC values were rounded up to an integer value for further analysis.

#### 2.4.1. Relationship between Baits, Number of *Culicoides* Sweep-Netted, and Distance

A GLMM with a Poisson distribution was fitted to explain both the daily numbers of *Culicoides* sweep-netted and the rate of blood-fed *Culicoides* sweep-netted within this daily sampling period [25]. The explanatory (fixed) variables assessed were the type of bait used to attract the midges, the distance in meters from these baits, and the interaction between bait and distance. Distance was modelled as a linear variable or by including natural cubic splines with a knot at 10 m. When modelling the rate of blood-fed *Culicoides*, the daily number of *Culicoides* caught was introduced as an offset variable.

Random effects were included to account for variations introduced by the experimental procedures. The assessed random effects were:(a)Day of sampling, which accounted for the influence of environmental variation (temperature and humidity) between sampling days and repeated measurements taken on the same hosts. Temperature and humidity were also assessed, but their influence was intrinsically explained by using day of sampling, leading to better model fits.(b)Sampler, i.e., the person performing the sweep-netting at different sampling locations (bait, distance); this accounts for variations introduced by different people performing the catches.(c)The observation effect, which accounts for daily varying overdispersion between individual *Culicoides* catches. This random effect was included only if the model needed correction for overdispersion.(d)Location, which assessed variation introduced by the samplings locations (which was evenly allocated to all hosts).

#### 2.4.2. Relationship between the Rate of Blood-Fed *Culicoides* Species and Distance

To model this relationship, GLMM Poisson models were fitted for each of the different baits. In this model, the number of blood-fed *Culicoides* was the response variable, the daily number of *Culicoides* caught was introduced as an offset variable, and *Culicoides* species, distance (linear or with natural cubic splines with a knot at 10 m), and the interaction between *Culicoides* species and distance were the explanatory variables. All of the random effects explained above were also assessed for these models.

## 3. Results

### 3.1. Distribution of Culicoides Midges

The total numbers of female *Culicoides* midges collected at a distance of 0 m were as follows: 7695 (cow), 533 (sheep), and 145 (LT) (Table 1). With increasing distance from the cow, the number of *Culicoides* midges caught decreased sharply. In contrast, the number of *Culicoides* midges caught at a distance of 10 m from the sheep was comparable to that at 0 m, but at a distance of 25 m, the number of *Culicoides* midges was clearly lower compared to the numbers caught at 0 and 10 m. The situation for the LT was quite different: increasing numbers of *Culicoides* midges were caught with an increasing distance to the LT. The total number of male *Culicoides* midges collected in the study for the major species were as follows: 3 (*C. obsoletus* complex), 1 (*C. chiopterus*), 0 (*C. dewulfi*), 0 (*C. punctatus*), and 0 (*C. pulicaris*) for the cow; 0 (*C. obsoletus* complex), 0 (*C. chiopterus*), 0 (*C. dewulfi*), 0 (*C. punctatus*), and 0 (*C. pulicaris*) for the sheep; and 3 (*C. obsoletus* complex), 2 (*C. chiopterus*), 34 (*C. dewulfi*), 7 (*C. punctatus*), and 13 (*C. pulicaris*) for the LT. *Culicoides* species richness at different distances from the cow was comparable: 15, 14, and 14 different species at 0, 10, and 25 m distance, respectively. In contrast, species richness at 0 m distance from the sheep was half of that at 10 and 25 m distance: 7, 15, and 15 different species at 0, 10, and 25 m distance, respectively. Remarkably, *Culicoides* species richness declined with increasing distances to the LT: 17, 12 and 11 different species at 0, 10 and 25 m distance, respectively. At 0 m distance, the proportion of blood-fed *Culicoides* midges was 7.6%, 6.6%, and 3.2% for the cow, sheep, and LT, respectively. The top-four most abundant *Culicoides* species—*C. obsoletus*, *C. dewulfi*, *C. chiopterus*, and *C. punctatus* (comprising 90–95% of all the *Culicoides* midges collected)—were detected equally in the environment of all the baits.

### 3.2. Culicoides Abundance and Bait and Distance Relationship

Based on GLMM modelling, the relationship between the mean number of *Culicoides* midges by bait and distance to bait is visualized in Figure 2. The random effects that remained in the models were the daily sources of variation due to the influence of weather (temperature and humidity) and the daily repeated measurements on the same baits, in addition to the daily varying overdispersion (observation effects) on the *Culicoides* distribution (Appendix A). In the presence of these two random effects, the contribution of the sampler to variations was negligible (i.e., close to zero). The location of the baits made no contribution to variations in the model and was found to be insignificant when assessed as a fixed effect.

#### 3.2.1. Distribution of *Culicoides* Midges by Sampling Hours

The mean number of *Culicoides* midges caught at five hours before sunset was almost nil; however, this value steadily increased to a peak at around one hour before sunset and then dropped to almost zero again at one hour after sunset for all baits and at all the distances investigated (Figure 3), covering a complete-as-possible measurement window in the evening. The number of blood-fed *Culicoides* midges collected by sweep-net steadily increased from almost zero at five hours before sunset to a peak at sunset before dropping to almost nothing again at one hour after sunset for the cow and the sheep and at all the distances investigated (Figure 4). The number of blood-fed *Culicoides* midges collected by sweep-net in the environment around the LT was negligible.

#### 3.2.2. Distribution of *Culicoides* Midges by Distance to Bait

The mean number of *Culicoides* midges sweep-netted at a distance of 10 m and 25 m distance from the cow was significantly (*p* < 0.001) lower, i.e., 6.5 times (95% confidence interval (CI): 4.3–9.7) and 14.7 times (95% CI: 9.8–22.2), compared to the mean number of *Culicoides* midges sweep-netted at a distance of 0 m, respectively (Figure 5). The mean number of *Culicoides* midges sweep-netted at a distance of 10 m from the sheep, however, was not significantly lower (Odds ratio = 1.02, 95% CI: 0.7–1.6) compared to the mean number of *Culicoides* midges sweep-netted at a distance of 0 m; the mean number of *Culicoides* midges sweep-netted at a distance of 25 m from the sheep was significantly lower (*p* = 0.005), i.e., 1.9 times (95% CI: 1.2–2.9), compared to the mean number of *Culicoides* midges sweep-netted at a distance of 0 m. The mean number of *Culicoides* midges sweep-netted at distances of 10 m and 25 m from the LT was significantly (*p* < 0.001) higher, i.e., 2.2 times (95% CI: 1.4–3.5) and 3.4 times (95% CI: 2.1–5.3), compared to the mean number of *Culicoides* midges sweep-netted at a distance of 0 m, respectively.

#### 3.2.3. Distribution of *Culicoides* Midges by Distance between Baits

The mean number of *Culicoides* midges sweep-netted at distances of 0 m, 10 m, and 25 m from a cow was 13.2 times (95% CI: 8.8–19.8), 2.1 times (95% CI: 1.4–3.2), and 1.7 times (95% CI: 1.1–2.6) higher than that measured at the same distances from a sheep. The mean number of *Culicoides* midges sweep-netted at a distance of 0 m and 10 m from a cow was 57.5 times (95% CI: 36.9–89.7) and 4.1 times (95% CI: 2.7–6.2) higher than that around a LT, respectively. At a distance of 25 m from a cow and the LT, the mean number of *Culicoides* midges sweep-netted was not significantly different (OR = 1.2, 95% CI: 0.8–1.8).

#### 3.2.4. Rate of Blood-Fed *Culicoides* Species and Distance

Blood-fed rates were highest at 0 m distance and declined significantly (*p* < 0.001) with increasing distance. No significant differences in the blood-fed rates of *Culicoides* species were seen between measurements around different baits. In addition, there were no significant differences in blood-fed rates among *Culicoides* species. No significant interactions were observed between distance and *Culicoides* species with respect to blood-fed rate. The random effects that remained in the model were the daily sources of variation and the sampler (Appendix A).

## 4. Discussion

In a randomized trial conducted over 12 days, we quantified—on a micro landscape level—the distance dependency of *Culicoides* midge abundance in a pasture with different baits. An obvious conclusion is that *Culicoides* midge abundance under field conditions is not homogeneously distributed in the micro landscape. The non-homogenous distribution stems from the need of the midges for a blood meal, leading them to track and aggregate around warm-blooded hosts, prior to landing and feeding on them. In our study, *Culicoides* midge abundance clearly decreased as distance to livestock hosts increased, i.e., a sharp decrease with dairy cows and a moderate decrease with sheep. This pattern clearly reflects the attraction of *Culicoides* midges to these livestock hosts. Another evident inference is that a dairy cow is a far stronger attractor of *Culicoides* midges than a sheep, which supports the findings of recent field studies [8,18,26], in which *Culicoides* midges were collected by direct aspiration from the skin of the livestock hosts. This seems logical, considering the difference in size of the animals and the correlating difference in the quantity of exhaled volatiles that act as attractants. Using Kleiber’s allometric scaling (i.e., weight to the power of 0.75) [17] to calibrate for a theoretical equal basal metabolism (Kleiber’s allometric: 143.3 (cow), 14.3 (sheep)), there was about a 10-times difference between the cow and the sheep, which approximately reflects the observed difference in the total number of *Culicoides* midges collected at 0 m distance from both livestock hosts. The observation that comparable numbers of *Culicoides* midges were sweep-netted at 0 and 10 m distance from a sheep may indicate that the level of attraction of *Culicoides* midges to a sheep is quite modest. In addition, the fact that the numbers *Culicoides* midges sweep-netted at 25 m distance from the cow, sheep, and LT were virtually identical is a good indication that the method used and the design of this study were valid and that at this distance, there is virtually no level of attraction by the baits.

The reliance upon LT data as a vector-to-host ratio proxy is based on the assumption that the number and species of midges caught in a LT equates to that lured by the animal host [27]. However, it has become evident that the LT performs far more poorly than previously supposed [8,10,28]. Furthermore, it seems that the midges aggregating around the LT are disoriented by the UV-light and not really lured by it [10]. In contrast to the *Culicoides* species collected in the vicinity of livestock hosts, ornothophilic *Culicoides* species like *C. festivipennis*, *C. kibunensis*, *C. pictipennis*, and *C. circumscriptum* [29,30] were collected predominantly by the LT.

Manual sweep-netting is seldom used as a method to collect *Culicoides* midges. We know of several studies that used a sweep-net mounted on a truck to collect *Culicoides* midges [31,32,33,34]. In the Netherlands, sweep-netting in early summer revealed *Culicoides* midges to be active at least a few hours before nightfall, when the LT is not yet effective in collecting midges [8]. A recent host preference experiment with a dairy cow, a Shetland pony, and a sheep, using direct aspiration to collect midges from the skin of the animal hosts, revealed *C. obsoletus* complex to be largely nocturnal, while *C. chiopterus* and *C. dewulfi* were found to be predominantly diurnal [9]. The sweep-net results in this study mirror those obtained using direct aspiration during the same hours of collection in the evening. Since there is good correlation between direct aspiration (pootering) and sweep-netting results [9], and given that sweep-netting is much easier and simpler to execute than pootering, the use of a sweep-net for collecting information pertaining to the vector/host ratio and host preference may be revived.

To estimate the blood-feeding rate, direct-aspiration (pootering) from the skin of hosts is preferable to sweep-netting, since pootering is the most direct measuring method to get an impression of the blood-feeding activity for a specific host. The proportion of blood-fed midges in our study using sweep-netting was approximately 5–10% for the cow and the sheep, while in earlier, comparable studies that used pootering to collect *Culicoides* midges, the proportion of blood-fed midges was around 20–30% [8,26,35,36].

Little is known about the fine-scale distribution of *Culicoides* midges in the landscape. Predominantly, LTs are used for *Culicoides* surveillance; LT-derived data are widely available and are therefore used to model vector distributions and the transmission of vector-borne diseases [4,37,38]. Working with these data and models has led to the identification of major knowledge gaps, in particular in terms of the translation of LT catches to midge densities [27]. Rigot et al. [14] indicated that a Leptokurtic model—with *Culicoides* abundance as a negative logarithmic function of the distance from a farm—fitted their data best. Our data mirror this phenomenon with respect to the distance dependency, but on an even finer scale, indicating that such a relationship differs depending on the host species and the distance from hosts, both of which have a profound effect on the actual *Culicoides* abundance in the landscape.

## Figures and Tables

**Figure 1 insects-14-00637-f001:**
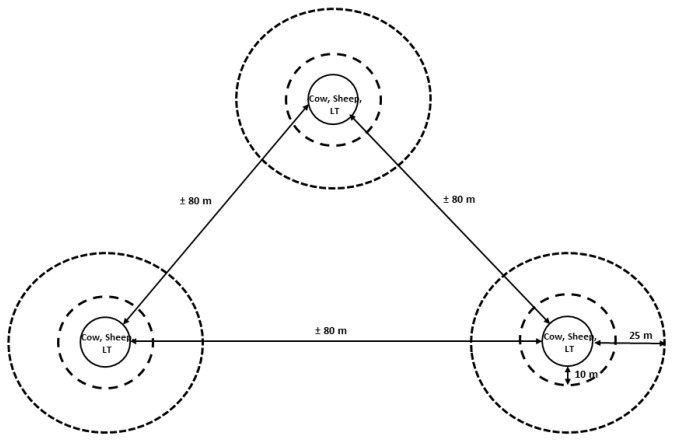
Diagram of the study design. A cow, a sheep, and a black-light suction trap (LT) used as baits were rotated between three different locations (with a distance of approximately 80 m between the locations, set in a triangle-shape in the landscape) in the pasture each observation day. The baits were sweep-netted at three different distances: 0, 10, and 25 m.

**Figure 2 insects-14-00637-f002:**
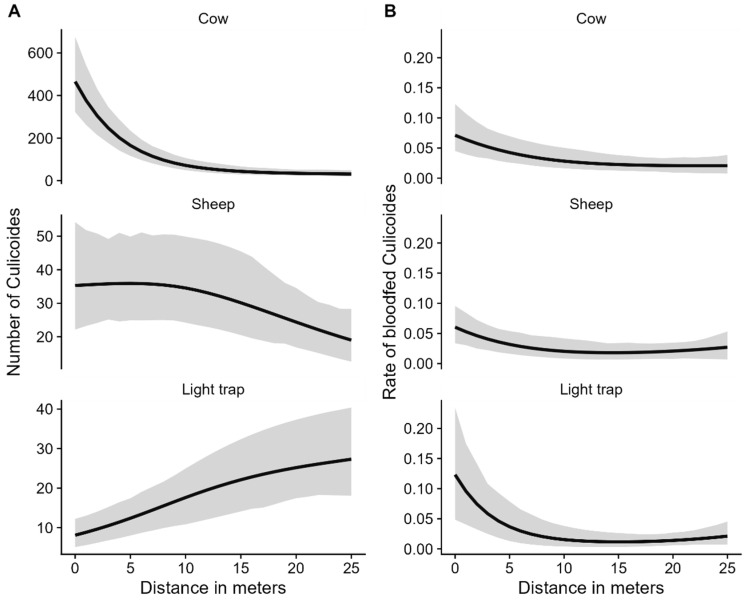
Mean predicted number of female *Culicoides* midges (with accompanying 95% confidence interval) by bait and distance to bait (Figure **A**) and mean predicted rate of blood-fed female *Culicoides* (with accompanying 95% confidence interval) by bait and distance to bait (Figure **B**) based on 12 daily sessions of 3 min of sweep-netting each hour from five hours before sunset up to and including one hour after sunset in the May–June period.

**Figure 3 insects-14-00637-f003:**
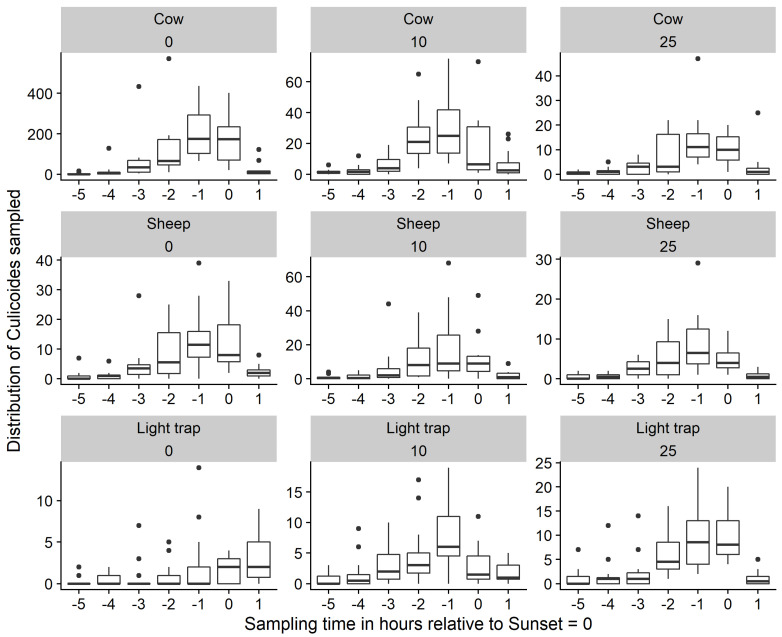
Distribution of sweep-net-collected female *Culicoides* midges by sampling hour, bait, and 0, 10, and 25 m distance from the bait (12 daily sessions of 3 min of sweep-netting each hour from five hours before sunset up to and including one hour after sunset in the May–June period; fat dark line in the box: median; lower end of the box: 25% quantile; higher end of the box: 75% quantile; highest bullet or high end of the vertical line coming out of the box: highest value; lowest bullet or low end of the vertical line coming out of the box: lowest value).

**Figure 4 insects-14-00637-f004:**
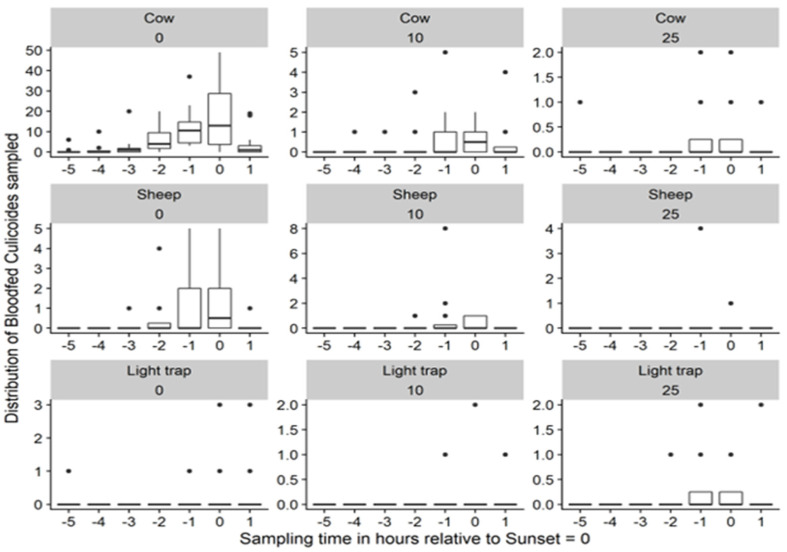
Distribution of sweep-net-collected blood-fed female *Culicoides* midges by sampling hour, bait, and 0, 10, and 25 m distance to the bait (12 daily sessions of 3 min of sweep-netting each hour from five hours before sunset up to and including one hour after sunset in the May–June period; fat dark line in the box: median; lower end of the box: 25% quantile; higher end of the box: 75% quantile; highest bullet or high end of the vertical line coming out of the box: highest value; lowest bullet or low end of the vertical line coming out of the box: lowest value).

**Figure 5 insects-14-00637-f005:**
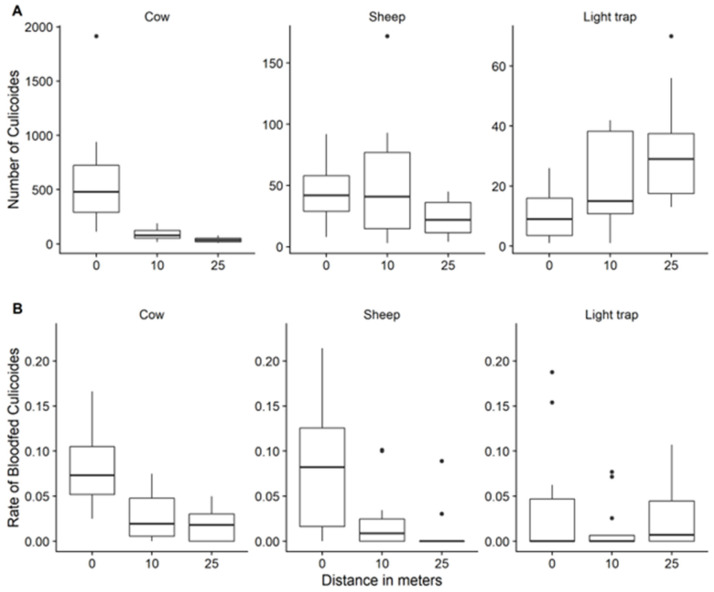
Distribution of sweep-net collected female *Culicoides* midges (Figure **A**) and rate of blood-fed female *Culicoides* midges (Figure **B**) for each type of bait between distances (12 daily sessions of 3 min of sweep-netting each hour from five hours before sunset up to and including one hour after sunset in the May–June period; fat dark line in the box: median; lower end of the box: 25% quantile; higher end of the box: 75% quantile; highest bullet or high end of the vertical line coming out of the box: highest value; lowest bullet or low end of the vertical line coming out of the box: lowest value).

**Table 1 insects-14-00637-t001:** Total number of female *Culicoides* midges at 0, 10, and 25 m distance from a cow, a sheep, and a black-light suction trap (collected over 12 daily sessions by 3 min of sweep-netting each hour from five hours before sunset up to and including one hour after sunset in the May–June period; ^#^ sweep-net result + (black-light suction trap result/20)).

Host Distance to Host Species	Cow			Sheep			Black-Light Suction Trap		
	0 m	10 m	25 m	0 m	10 m	25 m	0 m	10 m	25 m
	Total (%)	Total (%)	Total (%)	Total (%)	Total (%)	Total (%)	Total (%) ^#^	Total (%)	Total (%)
*C. obsoletus/scoticus*	2525 (32.8)	384 (35.3)	141 (28.7)	221 (41.5)	205 (32.1)	74 (25.0)	31.3 (21.5)	81 (26.3)	89 (23.0)
*C. dewulfi*	2813 (36.6)	226 (20.8)	88 (17.9)	206 (38.6)	160 (25.1)	66 (22.3)	31 (21.3)	70 (22.7)	91 (23.5)
*C. chiopterus*	1625 (21.1)	351 (32.2)	202 (41.1)	77 (14.5)	219 (34.3)	116 (39.2)	58.9 (40.5)	131 (42.5)	159 (41.1)
*C. punctatus*	327 (4.2)	40 (3.7)	24 (4.9)	15 (2.8)	19 (3.0)	7 (2.4)	12.8 (8.8)	16 (5.2)	20 (5.2)
*C. achrayi*	201 (2.6)	36 (3.3)	6 (1.2)	2 0.4)	3 (0.5)	10 (3.4)	2.4 (1.7)	2 (0.6)	9 (2.3)
*C. pulicaris*	90 (1.2)	15 (1.4)	7 (1.4)	4 (0.8)	10 (1.6)	2 (0.7)	3.4 (2.3)	2 (0.6)	8 (2.1)
*C. pallidicornis*	50 (0.6)	17 (1.6)	2 (0.04)	0	2 (0.3)	2 (0.7)	1.2 (0.8)	0	3 (0.8)
*C. impunctatus*	30 (0.4)	7 (0.6)	7 (1.4)	8 (1.5)	2 (0.3)	2 (0.7)	1.3 (0.9)	1 (0.3)	1 (0.3)
*C. heliophilus*	11 (0.1)	0	0	0	0	2 (0.7)	0	0	0
*C. stigma*	9 (0.1)	2 (0.2)	1 (0.02)	0	1 (0.2)	2 (0.7)	0.1 (0.03)	1 (0.3)	0
*C. pictipennis*	4 (0.05)	2 (0.2)	8 (1.6)	0	6 (0.9)	2 (0.7)	0.8 (0.6)	1 (0.3)	4 (1.0)
*C. albicans*	4 (0.05)	2 (0.2)	2 (0.04)	0	2 (0.3)	2 (0.7)	0.2 (0.1)	1 (0.3)	0
*C. sp.nr.newsteadi*	2 (0.03)	0	0	0	0	0	0	0	0
*C. grisescens*	2 (0.03)	0	0	0	0	0	0	0	0
*C. furcillatus*	2 (0.03)	4 (0.4)	2 (0.04)	0	0	6 (2.0)	0	0	2 (0.5)
*C. subfascipennis*	0	2 (0.2)	1 (0.02)	0	1 (0.2)	2 (0.7)	0.4 (0.3)	1 (0.3)	0
*C. festivipennis*	0	0	0	0	4 (0.6)	0	1.2 (0.8)	0	0
*C. lupicaris*	0	0	0	0	2 (0.3)	0	0.05 (0.03)	1 (0.3)	0
*C. segnis*	0	1 (0.1)	1 (0.02)	0	2 (0.3)	1 (0.3)	0	0	0
*C. kubenensis*	0	0	0	0	0	0	0	0	1 (0.3)
*C. riethi*	0	0	0	0	0	0	0.15 (0.1)	0	0
*C. fascipennis*	0	0	0	0	0	0	0.05 (0.03)	0	0
*C. circumscriptus*	0	0	0	0	0	0	0.2 (0.1)	0	0
Total (%)	7695 (100)	1089 (100)	492 (100)	533 (100)	638 (100)	296 (100)	145.4 (100)	308 (100)	387 (100)
Species richness	15	14	14	7	15	15	17	12	11

## Data Availability

Data collected in this study are available upon reasonable request to the corresponding author.

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
