# Peer review of "Culicoides* (Diptera: Ceratopogonidae) Abundance Is Influenced by Livestock Host Species and Distance to Hosts at the Micro Landscape Scale"

_insects, 2023, doi:10.3390/insects14070637_

Round 1

Reviewer 1 Report

Comments on Elbers and Gonzales “Culicoides abundance…”

Reviewed for Insects 9 June 2023

In the present study the authors have used humans conducting sweep netting at set distances (0, 10, 25m) from a cow, a sheep, and a light trap in order to assess levels of Culicoides activity. The light trap used was the Onderstepoort model, a highly effective and attractive suction trap with a strong UV light and fan relative to most other light trap models. This sampling was conducted starting well before sunset and continuing until an hour after sunset. The purpose was to compare the live hosts to each other and the trap over time, to supply a critical analysis of abundance. Over the 12 study nights, a large number and significant diversity of species were collected. Using sweep nets in this way is fairly novel in the literature, actually. It reminded me somewhat of very old literature (1960’s I think) in the Caribbean islands, where the author (Davies, if memory serves) used a sticky “bat” waved near the human body to try to quantify midge attack rates. While the method also lacks some of the standardization possible using other methods (such as timed use of enclosure nets) and imposes some level of human interference, I welcome this type of study, provided its potential drawbacks are acknowledged. After all, ALL sampling methods are biased in some way, by such things as our choice of technique, time and place of use, etc.

Unfortunately at the moment I cannot access my literature very well, and the Netherlands studies did produce several past studies on relative collections of Culicoides from animal species and traps. However, as I recall they did not use the sweep netting technique. The sweep net findings are mostly as one would expect from many other studies comparing animals and a light trap (including some done by the present authors), with the interesting exception of the light trap (as mentioned below), but more such studies are always helpful, and the sweep net methodology is sufficiently new to justify publication in Insects I think.

In general I agree that the triangular trap location design and 80m separation should have been adequate to minimize interference between the two animals and the trap. Rather than completely randomizing their location assignments, I would have preferred to have them, ultimately, evenly represented in each location (a bit in the style of a Latin Square design), but what they have done is acceptable.

The sweep netting should necessarily be described in quite a bit of detail, especially since the approach is not common and the researchers are necessarily imposing humans (with their attractive odors like CO2) into the sampling regime. The authors have done a nice job overall of explaining this, but a few questions remain. What mesh size was the net? It must of course have been unusually small to retain Culicoides. They mention back and forth sweeps. Is this approximately a 180 degree arc in front of the collector, who is walking continuously? What speed was the walking? This matters because a very slow pace increases the opportunity for collector interference. What they are trying to achieve is using the human sweep netting as an unbiased air sampler, hoping to minimize the attractiveness of the human themselves on the collections.

I object strenuously to the authors referring to the light trap as a “surrogate host”. In a recent review (McDermott and Mullens 2017, J. Med. Entomol.) the authors of the earlier paper go to some length to review the MANY ways this way of thinking is in error and they expand on it by explaining the conceptual risks of thinking or communicating in that way from the perspective of assessing epidemiological risks for vector-borne pathogens. The cues that attract Culicoides to a light trap (at least one that lacks added specific host kairomone cues like CO2) are probably totally different from those aspects of animals that attract blood feeding insects. The authors of the present paper clearly understand that, I think, but they need to banish this idea and terminology from their paper. A light trap is NOT in any way a surrogate for a host animal. Abundance may be well correlated sometimes between light traps and animal collections, but it is dangerous to specifically link them by calling a trap a surrogate host. Rather, a light trap is merely another useful collecting tool, using a different set of insect perception cues that we don’t actually grasp very well yet. So the authors need to go through the paper and change their nomenclature. If they feel driven to refer to the three things (cow, sheep and trap) collectively, they might say something like “baits” or “attractants”.

It would be nice if the authors had some bit of supportive data they could share on the relative abundance of obsoletus vs scoticus, perhaps from examination of males or molecular data. For example, if obsoletus seems to be far more common at the location than scoticus, a reader would be able to assume the data essentially apply to obsoletus.  But I understand this simply is not possible to do routinely with their collections.

Did the authors use any kind of subsampling to get through the collections? The numbers were significant, but did not absolutely require some kind of subsampling. At the beginning of Results section 3.1, the total numbers of Culicoides taken by the three baits needs to be restated somehow. At first reading it appears to be over 7 million, which of course was not what the authors intended and is cleared up be looking at Table 1. Perhaps they could do something like “The total Culicoides collected for the study were as follows: Cow (7,695), sheep (533) and light trap (145).” Speaking of males, perhaps I missed it, but were male and female Culicoides data lumped in the analysis (especially the light trap)? How common were males?

I was most intrigued with the fact that the number of Culicoides in the sweep netting actually declined as the human moved closer to the light trap. This might reflect the use of such a powerful light (and fan) as the Onderstepoort trap has, which as the authors suggested was quickly and permanently removing insects anywhere near it. Yet the authors’ suggestion that the trap may actually have confused the insects is somewhat supported by the surprisingly small number of insects actually caught in the trap. In that sense it might have been a good idea to use some type of trap that wasn’t quite so powerful! But it is too late to do that now, of course.

Here are a few more minor suggestions.

1)The Y-axis label in Fig. 2 is misspelled.

2)In the discussion the authors might want to mention the approximate difference in expected CO2 output for the cow vs the sheep (or a person, for that matter, which may be about like a sheep in CO2 output), based on their weight. It might be in the range of 5-10x or perhaps more, and there have been studies showing that some (most, I expect) Culicoides respond positively to increasing levels of CO2. No doubt other odor differences between the two hosts exist, as well as differences in silhouette size etc. But the “universal kairomone” CO2 is one commonality.

3) There are also other good truck trapping studies; the American papers by Barnard and Akey are great examples.

4) For use of odds ratio with this same system, the Gerry et al. 2009 Spain study was notable.

5) The Culicoides community misses Rudy Meiswinkel, whose influence on this study obviously was large and probably quite critical. It was nice of the authors to acknowledge this. If I might be so bold as to suggest it, might it be possible still to put him on the paper as an author, even though he has died? There certainly is some precedence for this in the literature, although I am not at all familiar with the policy in use by the journal Insects. However, this of course is up to the authors.

With moderate revision I feel the paper will be acceptable.

Author Response

The sweep netting should necessarily be described in quite a bit of detail, especially since the approach is not common and the researchers are necessarily imposing humans (with their attractive odors like CO2) into the sampling regime. The authors have done a nice job overall of explaining this, but a few questions remain. What mesh size was the net? It must of course have been unusually small to retain Culicoides. They mention back and forth sweeps. Is this approximately a 180 degree arc in front of the collector, who is walking continuously? What speed was the walking? This matters because a very slow pace increases the opportunity for collector interference. What they are trying to achieve is using the human sweep netting as an unbiased air sampler, hoping to minimize the attractiveness of the human themselves on the collections.

Reply by authors: we have added text in lines 129-130 and 132-134 to give more details as requested by the reviewer (changes have been highlighted by a yellow marker).

I object strenuously to the authors referring to the light trap as a “surrogate host”. In a recent review (McDermott and Mullens 2017, J. Med. Entomol.) the authors of the earlier paper go to some length to review the MANY ways this way of thinking is in error and they expand on it by explaining the conceptual risks of thinking or communicating in that way from the perspective of assessing epidemiological risks for vector-borne pathogens. The cues that attract Culicoides to a light trap (at least one that lacks added specific host kairomone cues like CO2) are probably totally different from those aspects of animals that attract blood feeding insects. The authors of the present paper clearly understand that, I think, but they need to banish this idea and terminology from their paper. A light trap is NOT in any way a surrogate for a host animal. Abundance may be well correlated sometimes between light traps and animal collections, but it is dangerous to specifically link them by calling a trap a surrogate host. Rather, a light trap is merely another useful collecting tool, using a different set of insect perception cues that we don’t actually grasp very well yet. So the authors need to go through the paper and change their nomenclature. If they feel driven to refer to the three things (cow, sheep and trap) collectively, they might say something like “baits” or “attractants”.

Reply by authors: we are aware of the remarks of the reviewer. We have skipped the term “surrogate host” from the manuscript and have changed that into baits and attractants are suggested by the reviewer throughout the text (see highlighted with yellow marker)

It would be nice if the authors had some bit of supportive data they could share on the relative abundance of obsoletus vs scoticus, perhaps from examination of males or molecular data. For example, if obsoletus seems to be far more common at the location than scoticus, a reader would be able to assume the data essentially apply to obsoletus.  But I understand this simply is not possible to do routinely with their collections.

Reply by authors: as indicated by the reviewer himself, this was indeed not possible.

Did the authors use any kind of subsampling to get through the collections? The numbers were significant, but did not absolutely require some kind of subsampling. At the beginning of Results section 3.1, the total numbers of Culicoides taken by the three baits needs to be restated somehow. At first reading it appears to be over 7 million, which of course was not what the authors intended and is cleared up be looking at Table 1. Perhaps they could do something like “The total Culicoides collected for the study were as follows: Cow (7,695), sheep (533) and light trap (145).” Speaking of males, perhaps I missed it, but were male and female Culicoides data lumped in the analysis (especially the light trap)? How common were males?

Reply by authors: we have added some text to make this more clear, and also changed the particular sentence, see lines 151-152, 200-201, and lines 207-212.

I was most intrigued with the fact that the number of Culicoides in the sweep netting actually declined as the human moved closer to the light trap. This might reflect the use of such a powerful light (and fan) as the Onderstepoort trap has, which as the authors suggested was quickly and permanently removing insects anywhere near it. Yet the authors’ suggestion that the trap may actually have confused the insects is somewhat supported by the surprisingly small number of insects actually caught in the trap. In that sense it might have been a good idea to use some type of trap that wasn’t quite so powerful! But it is too late to do that now, of course.

Reply by authors: as indicated by the reviewer, indeed it is too late to do so.

Here are a few more minor suggestions.

1)The Y-axis label in Fig. 2 is misspelled.

Reply by authors: that was indeed a mistake overseen by ourselves, we have revised the Y-axis label, see Figure 3 (due to adding a new Figure 1, the original Figure 2 is now Figure 3).

2)In the discussion the authors might want to mention the approximate difference in expected CO2 output for the cow vs the sheep (or a person, for that matter, which may be about like a sheep in CO2 output), based on their weight. It might be in the range of 5-10x or perhaps more, and there have been studies showing that some (most, I expect) Culicoides respond positively to increasing levels of CO2. No doubt other odor differences between the two hosts exist, as well as differences in silhouette size etc. But the “universal kairomone” CO2 is one commonality.

Reply by authors: we have added text to explain this as suggested by the reviewer, see lines 365-369.

3) There are also other good truck trapping studies; the American papers by Barnard and Akey are great examples.

Reply by authors:  we have added some more references on this, thanks to the suggestion of the reviewer, see lines 384-386.

4) For use of odds ratio with this same system, the Gerry et al. 2009 Spain study was notable.

Reply by authors: we have added this reference, see line 402.

5) The Culicoides community misses Rudy Meiswinkel, whose influence on this study obviously was large and probably quite critical. It was nice of the authors to acknowledge this. If I might be so bold as to suggest it, might it be possible still to put him on the paper as an author, even though he has died? There certainly is some precedence for this in the literature, although I am not at all familiar with the policy in use by the journal Insects. However, this of course is up to the authors.

Reply by authors: we have made this request to the editorial secretariat; unfortunately, it is not possible to add a new (co-)author after first submission, it would have been a too big hassle to try to repair this afterwards. So, we keep it as originally phrased in the Acknowledgements.

Reviewer 2 Report

This study uses a randomised block design and sweep-net catches to evaluate how host species identity (sheep, cow, and light traps used as a 'surrogate' host) and the measurement distance to these hosts impact Culicoides abundance using sweep netting. The impact of host identity and distance to host on Culicoides abundance by sweep-netting and interactions between host and distance were estimated by generalized linear mixed models Authors find that at micro-environmental scales (25 m and less) that there was a heterogenous distribution of Culicoides, with larger numbers of Culicoides adjacent to cows (0m) as compared to at increasing distances (10 and 25 m) from them. The methods and results from this study are of importance to the literature because they evaluate how host species and fine-scale distance from particular host species could influence culicoides abundance and also shows that sweep netting may be a more efficient technique, probably better than light traps, for studying and comparing Culicoides abundance in a variety of field settings. This study also quantifies how light traps can under-estimate Culicoides abundance. Furthermore, this study also shows how the pattern of the relationship between distance and Culicoides abudance and distance to host source can differ depending on the host source.

It would be nice to see a bit more detail about the model structures and results for different models in the supplementary data. I did not see any supplemental data in terms of the glmm model results and I think that information should be included in supplementary materials for the readers. Also, I suggest a small diagram of the study design would also be useful. 

There are some minor grammar errors but I just focused on the science details- this is something that the journal's editorial staff could check. 

Author Response

It would be nice to see a bit more detail about the model structures and results for different models in the supplementary data. I did not see any supplemental data in terms of the glmm model results and I think that information should be included in supplementary materials for the readers. Also, I suggest a small diagram of the study design would also be useful.

Reply by authors: as suggested we have put model results into a supplementary file with Table S1 and Table S2 (as indicated in the manuscript in line 255, and line 349).

Furthermore, we have added a new Figure 1 (line 116) with a diagram of the study design.